# 3D-MRI evaluation of cartilage thickness changes and their location in the patellofemoral joint after open wedge high tibial osteotomy for knee osteoarthritis: A retrospective cohort study

Yusuke Fuchioka[1], Nobutake Ozeki[1], Hideyuki Koga [2], Tomomasa Nakamura[2], Yusuke Nakagawa[2], Masaki Amemiya[2], Asuka Asami[1], Hisako Katano [1], Ichiro Sekiya [1]*

1 Center for Stem Cells and Regenerative Medicine, Institute of Science Tokyo, Tokyo, Japan,
2 Department of Joint Surgery and Sports Medicine, Graduate School of Medical and Dental Sciences, Institute of Science Tokyo, Tokyo, Japan

* sekiya.arm@tmd.ac.jp

## Abstract

### Background

Open wedge high tibial osteotomy (OWHTO) has been widely established as a safe surgical procedure for medial compartmental knee osteoarthritis. However, concerns remain regarding the progression of patellofemoral (PF) osteoarthritis following surgery. Recent advances in 3D-MRI analysis have enabled quantitative cartilage thickness measurement. We hypothesized that OWHTO would result in measurable decreases in the PF joint cartilage thickness, predominantly medially and detectable using quantitative 3D-MRI. This study evaluated the clinical utility of quantitative 3D-MRI for assessing PF joint cartilage changes before and after OWHTO.

### Methods

Patients were included if they had undergone OWHTO without lateral retinacular release for medial osteoarthritis and had both the preoperative and post–hardware-removal 3D-MRI datasets required for this analysis. Radiographic evaluations were performed before and after OWHTO. Trochlear and patellar cartilage thicknesses were measured from 3D-MRI images at both time points. Changes exceeding 0.1 mm (the validated measurement precision threshold) were considered significant. To assess cartilage loss location, each 3D image was divided into medial, central, and lateral thirds. Superimposed images were observed to determine spatial correspondence of the cartilage defects.

**Data availability statement:** All relevant data are within the paper and its Supporting information files.

**Funding:** This study was funded by Institute of Science Tokyo.

**Competing interests:** The authors declare no competing interests.

**Abbreviations:** OWHTO, Open wedge high tibial osteotomy; OA, Osteoarthritis; PF, Patellofemoral; 3D-MRI, Three-dimensional magnetic resonance imaging; ROI, Region of interest; PDW, Proton density weighted; SPGR, Spoiled gradient recalled; BMI, Body mass index; MRI, Magnetic resonance imaging.

## Results

In total, 13 knees from 13 patients (median age 55 [32–74] years) were evaluated. Postoperatively, patellar height and lateral tilt significantly decreased (p < 0.001 for both). Of these 13 cases, 7 (54%) showed thickness reductions exceeding 0.1 mm in the trochlear cartilage, and 7 cases showed reductions in the patellar cartilage, with 4 cases showing reductions in both. All cases demonstrated predominantly medial thickness decreases (p = 0.008). Of the 3 cases with patellar cartilage defects, 2 cases showed spatial correspondence with trochlear defects.

## Conclusions

Quantitative 3D-MRI analysis revealed significant cartilage thickness decreases after OWHTO, predominantly in the medial aspect of the PF joint. This method proved useful for evaluating postoperative PF joint changes and detecting cartilage defect locations.

## Introduction

Open wedge high tibial osteotomy (OWHTO) is a well-established surgical procedure for the treatment of medial compartmental knee osteoarthritis (OA) with varus malalignment [1]. This procedure corrects lower limb alignment by redistributing the weight-bearing load from the medial to the lateral compartment. While OWHTO has demonstrated favorable clinical outcomes for unicompartmental OA, concerns have been raised regarding the potential progression of patellofemoral (PF) joint degeneration following this procedure [2], as the surgery can cause alterations in the PF joint biomechanics, including changes in patellar height and alignment [3]. However, accurate assessment of these postoperative PF joint changes has been challenging using conventional imaging methods.

In recent years, three-dimensional magnetic resonance imaging (3D-MRI) analysis has emerged as a powerful tool for the quantitative assessment of articular cartilage [4–6]. Technological advances in 3D-MRI have now enabled precise measurements of cartilage thickness and detailed evaluation of morphological changes in specific anatomical regions [7,8], including the PF joint [9]. Compared to conventional radiography or standard MRI, 3D-MRI provides comprehensive volumetric data that allow more accurate detection and monitoring of cartilage degeneration patterns [10]. Nevertheless, despite their potential to enhance our understanding of these alterations, these advanced imaging techniques have been underutilized in the examination of the details of post-OWHTO cartilage thickness changes.

Accurate quantification of cartilage thickness changes could provide valuable insights into the mechanisms of PF joint degeneration and could potentially guide surgical planning. Understanding these changes could enable surgeons to identify high-risk patients, optimize surgical techniques, and develop targeted rehabilitation protocols to minimize PF joint deterioration. We hypothesized that OWHTO would

result in measurable decreases in PF joint cartilage thickness, predominantly in the medial aspect, and that we could detect these changes using quantitative 3D-MRI. The aim of this study was to evaluate the clinical utility of quantitative 3D-MRI in assessing PF joint cartilage changes before and after OWHTO, with particular attention paid to the spatial distribution of cartilage wear and the relationship between patellar and trochlear cartilage changes.

## Methods

### Subjects

This study was conducted in accordance with the Declaration of Helsinki and was approved by our institutional review board. All participants provided written informed consent before surgery, granting permission for the use of their anonymized clinical data for research purposes.

### Patient selection and criteria

We retrospectively included patients from a single center who underwent opening wedge high tibial osteotomy (OWHTO) without lateral retinacular release of the PF joint for medial OA between 2015 and 2023. Cases with lateral retinacular release were excluded because the procedure can alter PF joint alignment and cartilage status. All included patients had analyzable 3D-MRI data obtained following a standardized imaging protocol within 3 months before surgery and within 3 months after implant removal.

#### Inclusion criteria:

- Patients aged 18 years or older

- Diagnosis of medial compartment knee OA

- Underwent OWHTO without lateral retinacular release of the PF joint

- 3D-MRI data available both preoperatively and after implant removal

- Minimum follow-up period of 1 year

#### Exclusion criteria:

- Previous knee surgery on the affected limb

- Incomplete or poor-quality MRI data, loss to follow-up

### Surgical procedure

The OWHTO procedure followed the method proposed by Staubli et al. [11]. The aim of the preoperative plan was to shift the weight-bearing line ratio to a point 57% laterally along the transverse diameter of the tibial plateau. This target was chosen based on the method described by Katagiri et al., in which centralization and preservation of the medial meniscus function lead to a correction closer to neutral alignment than is achieved using the conventional 62% setting [12]. First, an osteotomy was performed in the coronary plane beneath the tibial tuberosity. This was followed by a transverse osteotomy of the posterior tibia. The osteotomy site was opened to the planned correction width with intraoperative fluoroscopic assessment of the medial proximal tibial angle. Wedge-shaped β-tricalcium phosphate blocks (Osferion 60; Olympus Terumo Biomaterials, Tokyo, Japan) were placed into the osteotomy sites. The osteotomy site was stabilized using either Tris plates (Olympus Terumo Biomaterials, Tokyo, Japan) or Tomofix plates (DePuy Synthes Johnson & Johnson, Tokyo, Japan) with locking screws (Fig 1A). Approximately one year later, an arthroscopic examination was performed (Fig 1B) in conjunction with implant removal.

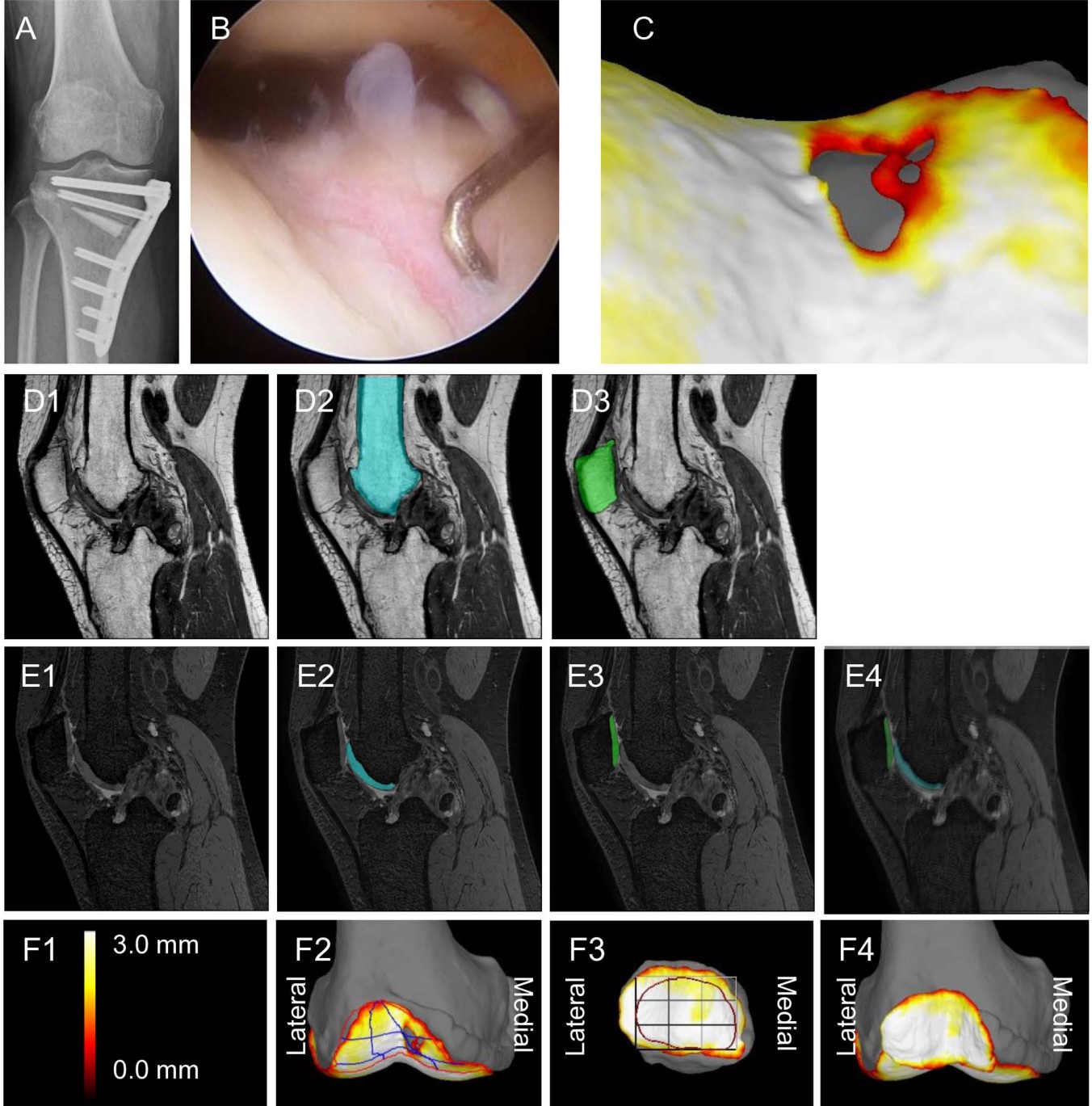

**Fig 1. An exemplary case one year after open wedge high tibial osteotomy (OWHTO) for right knee osteoarthritis. (A)** Radiographic image after OWHTO. **(B)** Arthroscopic image of the femoral trochlear cartilage at one year postoperatively. A cartilage defect is visible in the medial trochlea. **(C)** 3D-MRI image of the trochlea. Similarly, a defect is visible in the medial trochlea. **(D)** Sagittal PDW MRI: (1) Plain image. (2) Automated femoral bone extraction (blue). (3) Automated patellar bone extraction (green). **(E)** Sagittal SPGR MRI: (1) Plain image. (2) Automated femoral cartilage extraction (blue). (3) Automated patellar cartilage extraction (green). (4) Combined automated extraction of femoral cartilage (blue) and patellar cartilage (green). **(F)** Cartilage thickness mapping: (1) Color scale for cartilage thickness. (2) 3D-MRI of the femoral trochlea. A region of interest (ROI) is defined for cartilage thickness quantification, which is divided into 9 subregions. (3) 3D-MRI of the patella, horizontally flipped for better visualization when super-imposed on the femoral trochlea. The defined ROI is similarly divided into 9 subregions. (4) 3D-MRI of patellar cartilage superimposed on the femoral trochlear 3D-MRI.

## Radiological evaluation

Radiographic images of the knee joint were evaluated preoperatively and approximately one year postoperatively. The femorotibial angle and weight-bearing line ratio were measured using full-length double-leg standing radiographs [13]. The medial proximal tibial angle was measured on anteroposterior standing radiographs as the medial angle between the tibial anatomical axis and the tibial plateau. The posterior tibial slope was measured on lateral radiographs at full extension as the angle between the tibial plateau and the perpendicular line to the tibial anatomical axis. The Caton-Deschamps index, which is calculated as the ratio of the distance from the lower edge of the patellar articular surface to the anterosuperior border of the tibia divided by the length of the patellar articular surface, was measured on lateral radiographs in full extension [14,15]. Other measurements determined from axial radiographs included the lateral patella shift, expressed as the percentage of lateral displacement of the patella relative to the femoral trochlear groove, and the tilting angle, expressed as the angle between the posterior condylar line of the femur and the maximal width of the patella [16]. Although cartilage thickness cannot be directly assessed on radiographs with the same accuracy as it can on MRI, these radiographic measurements provide valuable information on lower limb alignment, patellar position, and other bony parameters that can influence PF joint mechanics. These parameters are essential for interpreting postoperative changes in cartilage condition in the context of overall knee biomechanics. All radiographic measurements were performed by a single experienced orthopedic surgeon (YF), and in cases of uncertainty, the final decision was made in consultation with another senior orthopedic surgeon (NO).

## Three-dimensional MRI analysis

The software for MRI analyses was a three-dimensional image analysis system volume analyzer (SYNAPSE 3D [Japanese product name: SYNAPSE VINCENT] collaborative version, FUJIFILM Corporation, Tokyo, Japan). The software provides cartilage thickness mapping by displaying cartilage thickness as a color scale, with thick areas represented in white and thin areas in red [10]. This system presents three-dimensional images of any location of interest (Fig 1C).

Proton density weighted (PDW) images were used for automatic segmentation of the bone region (Fig 1D), and spoiled gradient recalled (SPGR) images were used for automatic segmentation of the cartilage region (Fig 1E). The 3D image was then reconstructed (Fig 1F). The femoral cartilage was rotated so that the trochlear groove faced forward and was then projected onto a plane. The patellar cartilage was rotated and tilted to maximize the patellar cartilage area and projected onto a plane. This orientation was chosen because it provides the clearest view and the most comprehensive information for assessment, while also standardizing the measurements and reducing variability. The patellar cartilage was shown inverted medially and laterally in the 3D image, allowing the patellar cartilage to be overlapped onto the femoral cartilage [9].

The software automatically set the region of interest (ROI) of the femoral trochlea and patellar cartilage (Fig 1F). In this study, the patellar ROI was set at 2 mm inside the cartilage. The preoperative ROI was copied to the postoperative images. When the preoperative ROI was deemed inappropriate, manual corrections were made, and these cases were documented. An ROI was considered inappropriate when it was clearly misaligned with the actual femoral trochlea or patellar cartilage; this most commonly occurred in cases with large cartilage defects and markedly altered cartilage morphology.

The mean cartilage thickness in the ROI was quantified. Based on the analytical study by Katano et al., who examined inter-measurement errors of cartilage thickness, cases with cartilage thickness changes exceeding 0.1 mm were considered to be beyond the measurement error [17]. For these cases, although the ROIs of the femoral trochlea and patellar cartilage were automatically divided into 9 subregions (3 vertical × 3 horizontal), the ROIs were grouped into the medial 1/3, central 1/3, and lateral 1/3 by combining three vertical sections. The 3D images were then visually assessed to determine which of these three subregions had changed. This visual assessment was performed for all cases.

## Statistical analysis

Patient characteristics and radiological evaluation measurements were presented as medians (minimum–maximum). For cartilage thickness measurements by 3D-MRI analysis, both the actual values and the amounts of change were graphed using Excel (Microsoft Corporation, Redmond, WA, USA). Preoperative and postoperative comparisons of radiological evaluations were analyzed using the Wilcoxon signed-rank test. The binomial test was performed to determine whether cartilage changes were significantly shifted toward either the medial or the lateral side. Statistical analyses were performed using the BellCurve software for Excel (Social Survey Research Information Co., Ltd. Tokyo, Japan), with statistical significance set at p = 0.05.

## Results

### Characteristics of the patients

In total, 13 knees from 13 patients who underwent OWHTO without lateral retinacular release for medial OA between April 2015 and March 2023 also had both the preoperative and post–hardware-removal 3D-MRI datasets required for the present analysis (Fig 2). These patients were selected from a total of 387 knee osteotomy cases, of which 344 had undergone OWHTO. While 18 patients had analyzable 3D-MRI data before and after surgery, 5 were excluded due to concurrent lateral retinacular release. The final cohort of 13 patients consisted of 8 females and 5 males, with a median age of 55 (32–74) years. The median BMI was 23.9 (18.5–30.8) kg/m2, and the median interval between the OWHTO and the second MRI scan was 19 (10–62) weeks. Among the 13 cases, 10 had undergone medial meniscal procedures: one case with excision, three cases with repair, three cases with centralization, and three cases with combined excision and centralization.

### Preoperative and postoperative radiological evaluation

The osteotomy parameters, including the femorotibial angle, weight-bearing line ratio, medial proximal tibial angle, and posterior tibial slope, showed significant changes postoperatively, indicating improvement of the varus deformity (Table 1). The Caton-Deschamps index, which is an indicator of patellar height, showed significant decreases, demonstrating inferior

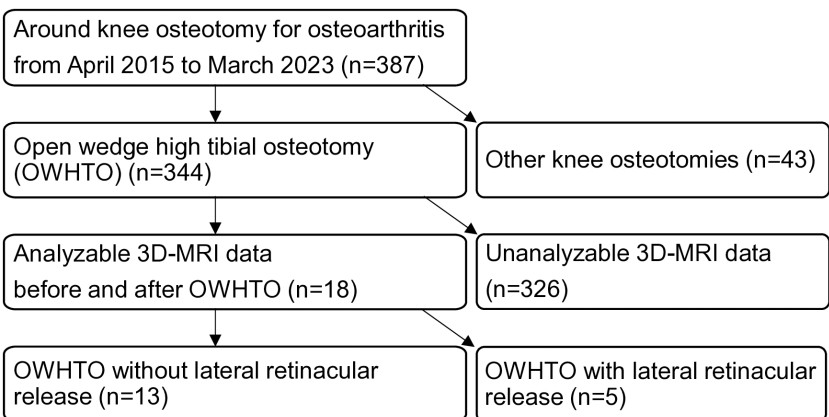

**Fig 2. Flow diagram of patient selection for the study.** From a total of 387 cases, 344 underwent OWHTO and 43 underwent other knee osteotomies. Of the OWHTO cases, only 18 had analyzable 3D-MRI data from both before and after surgery; the data from the other 326 cases were unanalyzable. The final analysis included 13 patients who underwent OWHTO without lateral retinacular release and 5 patients who underwent OWHTO with lateral retinacular release.

**Table 1. Pre- and postoperative radiological evaluations (n = 13).**

| | Before | After | p value |
|---|---|---|---|
| Femoro-tibial angle (degree) | 182.1 (177.2–184.9) | 172.9 (170.1–174.6) | *** |
| Weight bearing line ratio (%) | 19.4 (2.8–39.8) | 54.2 (44.6–69.4) | *** |
| Medial proximal tibial angle (degrees) | 84.0 (82.4–86.7) | 92.1(89.0–95.9) | *** |
| Posterior tibial slope (degrees) | 6.6 (4.6–12.6) | 8.6(2.9–16.0) | *** |
| Caton-Deschamps index | 0.84 (0.75–1.1) | 0.75 (0.54–1.05) | *** |
| Lateral patella shift (%) | 9.2 (2.9–11.7) | 9.6 (4.2–16.1) | *** |
| Tilting angle (degree) | 9.6 (4.6–12.8) | 5.4 (2.8–9.2) | *** |

Values are expressed as median (minimum-maximum).
***, p < 0.001 by Wilcoxon signed-rank test.

positioning of the patella postoperatively. The lateral patellar shift significantly increased, indicating lateral displacement of the patella. The patellar tilt significantly decreased, showing a postoperative reduction in the lateral tilt.

### Trochlear cartilage

The preoperative trochlear cartilage thickness ranged from 1.8 mm to 3.1 mm (Fig 3A). Postoperatively, that thickness decreased in 9 knees and increased in 4 knees. With the inter-measurement error threshold set at 0.1 mm, 7 cases (54%) showed a decrease exceeding this threshold. Upon observing the 3D images of these 7 cases, Knees 01–03 showed progression of cartilage wear and/or defects in the medial 2/3, while Knees 04–07 showed progression in the medial 1/3 (Fig 4). All seven knees with decreased cartilage thickness showed predominant and significant medial reduction, with no cases showing predominantly lateral reduction (p = 0.008, binomial test). One case (Knee 13) showed an increase exceeding 0.1 mm, with changes observed in the lateral 1/3. A review of this patient's MRI and radiographic data revealed no specific factors that could explain this change.

### Patellar cartilage

The preoperative patellar cartilage thickness ranged from 1.9 mm to 3.5 mm (Fig 3B). Postoperatively, that thickness decreased in 9 knees and increased in 4 knees. With the inter-measurement error threshold set at 0.1 mm, 7 cases showed a decrease exceeding this threshold. Upon observing the 3D images, Knees 01, 04, 07, and 10 showed progression of defects and/or wear in the medial 1/3, while Knees 05, 12, and 13 showed progression in the medial 2/3 (Fig 5). Statistical analysis revealed that cartilage reduction occurred predominantly in the medial region, with no cases showing primary lateral involvement (p = 0.008, binomial test). One case (Knee 08) showed an increase exceeding 0.1 mm, with changes apparent throughout the entire cartilage. No distinctive imaging findings were identified that could account for this observation.

### Patellar cartilage superimposed on trochlear cartilage

Three cases (Knees 01, 04, and 10) showed postoperative patellar cartilage defects in the 3D MRI images when the patellar cartilage was superimposed on the trochlear cartilage (Fig 6). In two of these cases (Knees 01 and 04), the locations of cartilage defects matched between the patellar and trochlear surfaces. These matching defects represented kissing lesions, which refer to opposing cartilage defects on articulating surfaces [18]. In all three cases, the findings suggest that contact with a medial trochlear osteophyte may have contributed to patellar cartilage wear.

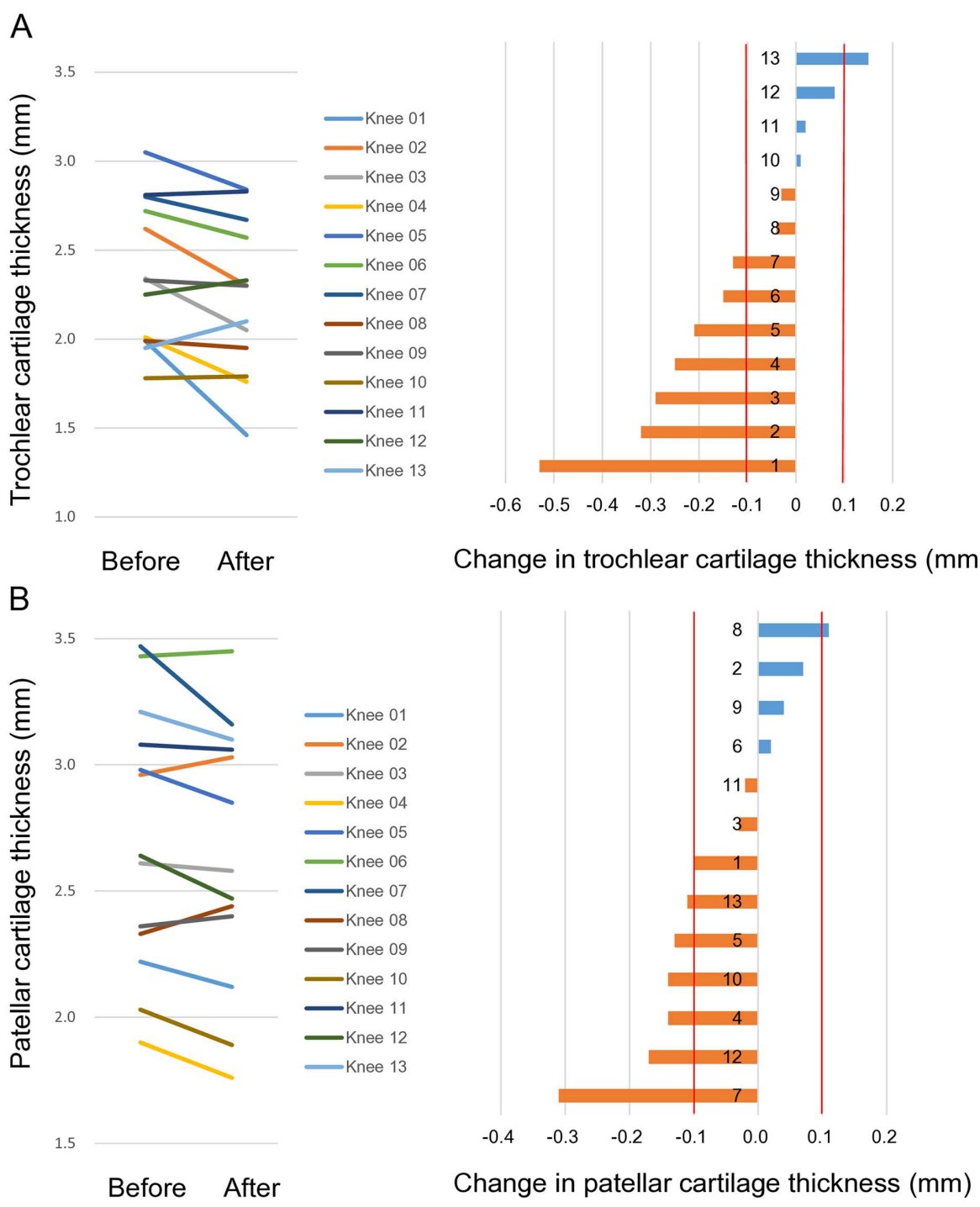

**Fig 3. Quantification of cartilage thickness in all cases. (A)** Preoperative and postoperative femoral trochlear cartilage thicknesses for each case and the thickness changes arranged in descending order. The average cartilage thickness was calculated from the overall ROI combining all 9 subregions. Red lines indicate inter-measurement error thresholds of −0.1 mm and 0.1 mm. **(B)** Preoperative and postoperative patellar cartilage thicknesses for each case and their changes arranged in descending order.

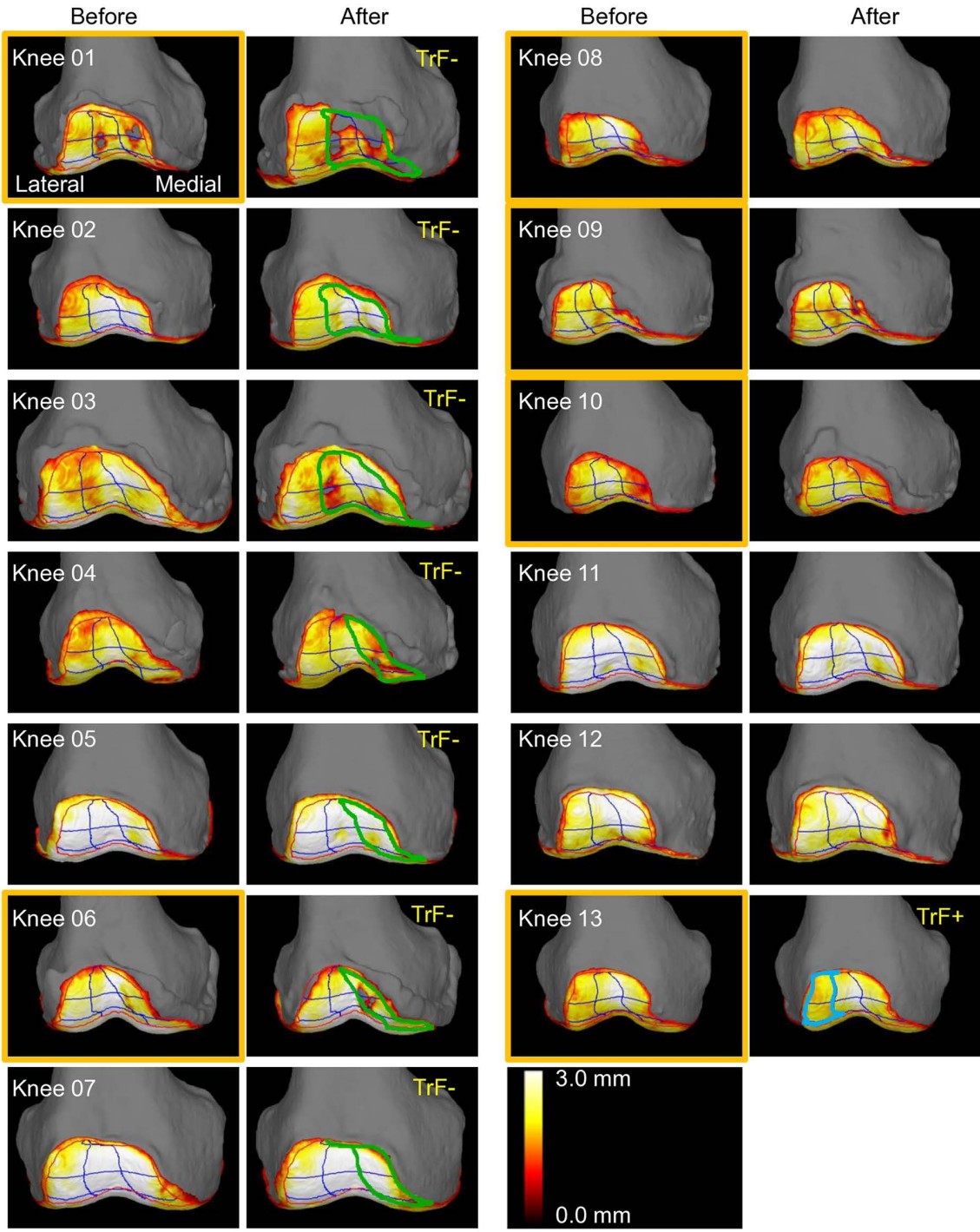

**Fig 4. Femoral trochlear 3D-MRI before and after OWHTO in all cases.** Images of the left knees are horizontally flipped for consistent visualization. Cases are numbered in order of the cartilage thickness reduction. Orange frames indicate cases where the ROI was manually modified. Cases with trochlear cartilage thickness reductions greater than 0.1 mm were labeled as TrF-, while cases with increases greater than 0.1 mm were labeled as TrF+. Among these, the nine ROIs were grouped into lateral, central, and medial thirds, and upon visual assessment, ROIs showing postoperative changes were outlined in green to indicate a reduction and blue to indicate an increase.

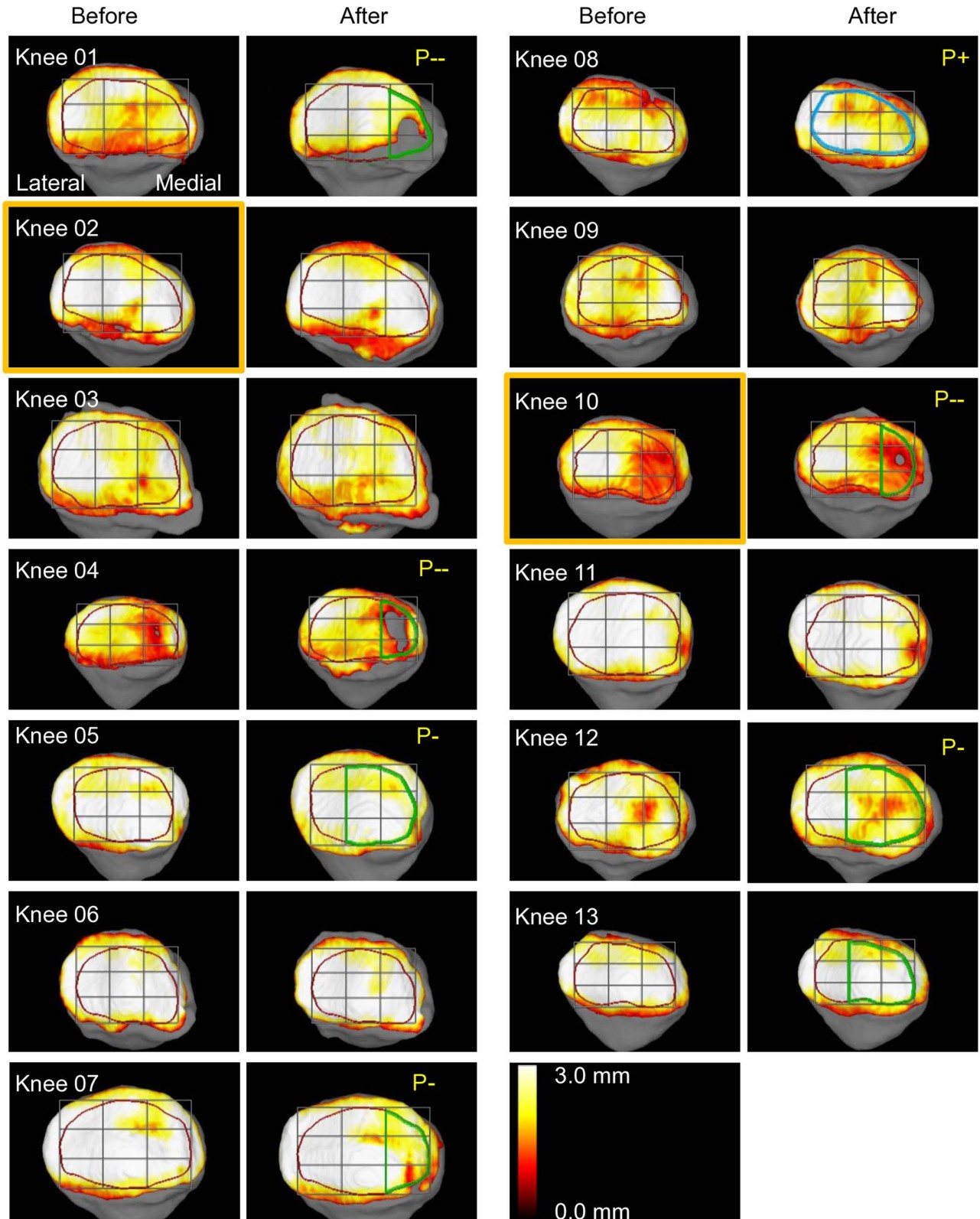

**Fig 5. Patellar 3D-MRI before and after OWHTO in all cases.** Images are oriented with the lateral side on the left and the medial side on the right to match the superimposition view of the trochlea. Cases are numbered in order of the cartilage thickness reduction observed in the femoral trochlear

3D-MRI. Orange frames indicate cases where the ROI was manually modified. Cases with patellar cartilage thickness reductions greater than 0.1 mm were labeled as P-, with those showing patellar cartilage defects were labeled as P--. Cases with increases greater than 0.1 mm were labeled as P+. Among these, the nine ROIs were grouped into lateral, central, and medial thirds, and upon visual assessment, ROIs showing postoperative changes were outlined in green to indicate a reduction and blue to indicate an increase.

## Detailed analysis

We examined Knee 01 in greater detail, as it was the case that showed the greatest decrease in femoral trochlear cartilage thickness. In the lateral view of the postoperative 3D image that combined the femoral trochlea and patella/patellar cartilage, the osteophyte at the inferior pole of the patella appeared to penetrate into the cartilage defect of the trochlea (Fig 7A). In the postoperative sagittal 2D MRI, the patella was positioned lower and tilted downward relative to the femur (Fig 7B). The oblique view centered on the osteophyte at the inferior pole of the patella confirmed that this osteophyte was in contact with the cartilage defect in the trochlear region (Fig 7C, 7D).

## Discussion

This study employed quantitative 3D-MRI analysis to investigate PF joint cartilage changes before and after OWHTO in patients with medial compartmental OA. Our findings demonstrated significant postoperative reductions in patellar height and lateral tilt. Cartilage thickness reduction exceeding 0.1 mm was observed in both the trochlear and patellar regions independently in 54% of the cases, with 31% showing reduction in both regions. All cases with cartilage wear demonstrated a distinct pattern of medial-dominant degradation. Some cases showed spatial correspondence between patellar and trochlear cartilage defects, suggesting the possibility that these opposing defects may develop through mechanical interaction, similar to "kissing lesions" [18].

The radiological findings revealed significant postoperative changes in PF joint alignment following OWHTO. While correction of the varus deformity was achieved, as evidenced by improvements in the femorotibial angle and mechanical axis, the procedure caused notable changes in the patellar positioning. The significant decrease in the Caton-Deschamps index indicates postoperative patella baja, in agreement with previous findings by Lee et al. [19]. This inferior positioning of the patella, combined with the increased lateral patellar shift and decreased lateral tilt, suggests an alteration in the biomechanics of the PF joint. Our findings partially agree with those from Gaasbeek's biomechanical study [20], which reported a decreased lateral tilt but no change in the lateral shift. These combined changes in the patellar position and alignment suggest that the significant alteration in the PF joint biomechanics following OWHTO could be a crucial factor determining subsequent joint degeneration.

Our selection of a threshold of 0.1 mm for detecting significant cartilage changes was carefully made based on rigorous validation studies of the measurement precision of 3D-MRI. Our previous research has established the reliability of this measurement system through multiple studies. For example, Sekiya et al. [21] demonstrated that the interscan measurement error was consistently below 0.10 mm across all knee regions in healthy volunteers. Furthermore, Katano et al. [22] validated the reproducibility of these measurements using systems from different MRI vendors and confirmed that 75% of the measurements had inter-measurement errors of ≤0.10 mm. A subsequent study on osteoarthritic knees demonstrated similar precision, with 67% of the measurements showing interscan errors of ≤0.10 mm [17]. Therefore, our observed cartilage thickness reductions exceeding 0.1 mm likely represent genuine tissue changes rather than measurement variability.

The predominant medial wear pattern in both the patellar and trochlear cartilages following OWHTO was a key finding that aligns with previous MRI studies using T2 mapping, as both Nakagawa et al. [23] and Komaki et al. [24] reported increased T2 values in the medial and central regions of the PF joint. Following OWHTO, the distal displacement of the tibial tubercle causes an increase in the tension in the extensor mechanism above that observed in the preoperative state.

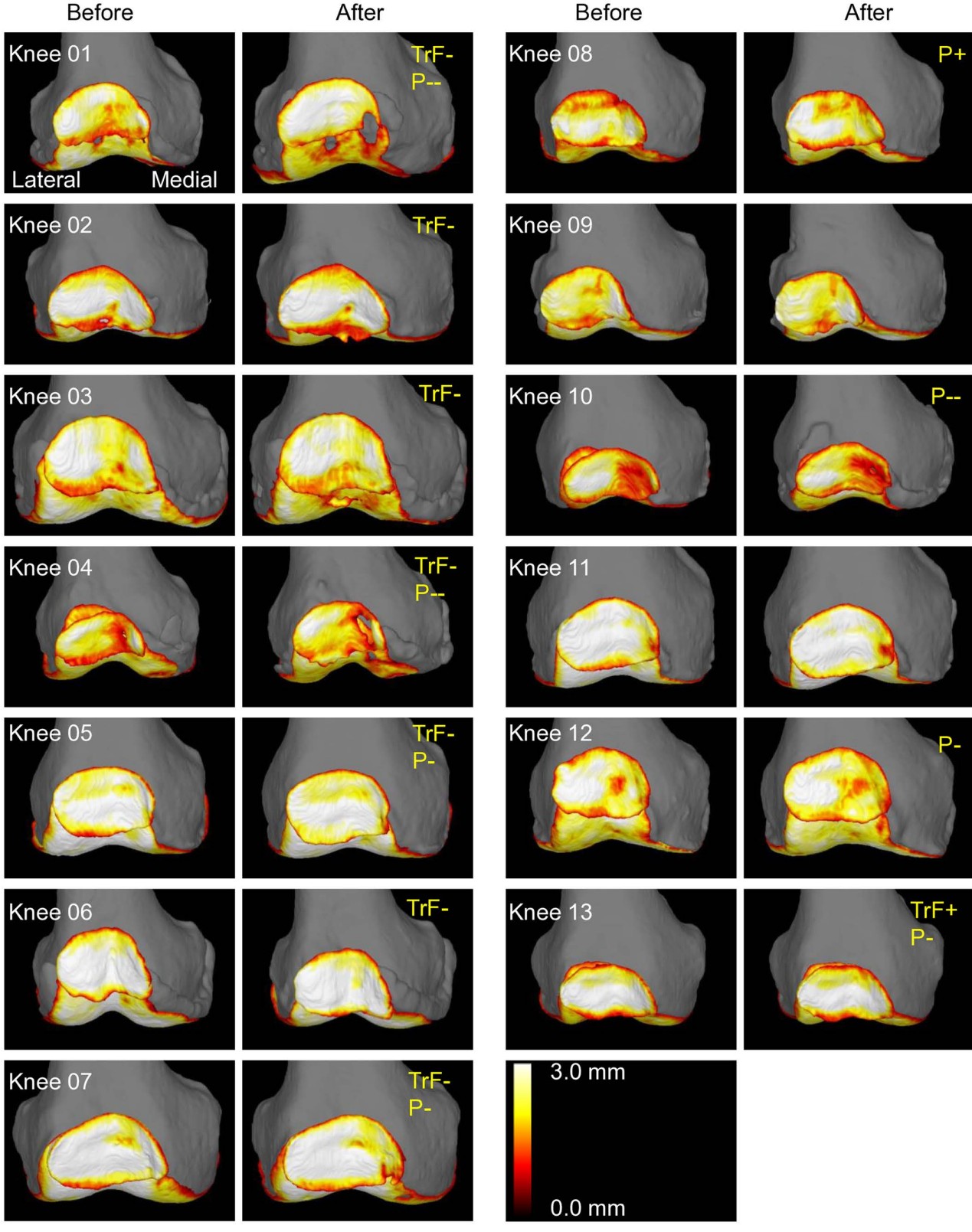

**Fig 6. Combined 3D-MRI showing the patellar cartilage superimposed on the femoral trochlear cartilage before and after OWHTO in all cases.** Images of left knees were horizontally flipped for consistent visualization. Cases are numbered in order of cartilage thickness reduction in femoral trochlear 3D-MRI. Cases with trochlear cartilage thickness reductions greater than 0.1 mm were labeled as TrF-, while cases with increases greater than 0.1 mm were labeled as TrF+. Cases with patellar cartilage thickness reductions greater than 0.1 mm were labeled as P-, with those showing patellar cartilage defects were labeled as P--. Cases with increases greater than 0.1 mm were labeled as P+.

Furthermore, the observed decrease in lateral patellar tilt alters the pressure distribution within the PF joint. This combination of patella baja and decreased lateral tilt appears to create elevated contact pressures, particularly on the medial aspects of both the patellar and trochlear cartilages.

Our spatial analysis of cartilage defects using superimposed 3D MRI images provided valuable insights into the pattern of PF joint degeneration after OWHTO. Of particular interest was the observation of kissing lesions, where the cartilage defects on the patellar surface corresponded spatially with the defects on the trochlear surface in two of the three cases that showed postoperative patellar cartilage defects. The presence of these kissing lesions strongly suggests that the cartilage degeneration follows a coordinated pattern of wear, likely resulting from the altered contact mechanics after OWHTO. This deduction is supported by the spatial correspondence of patellar and trochlear defects observed in multiple cases, indicating that degeneration on one surface occurs in regions directly opposing degeneration on the other. Similar matched defect patterns in the PF joint have been reported in studies of malalignment and altered PF kinematics, where increased localized contact stress was associated with accelerated cartilage loss [25,26]. These findings collectively support the interpretation that postoperative changes in PF alignment and biomechanics after OWHTO can create specific zones of elevated mechanical stress that can lead to the coordinated degeneration of opposing cartilage surfaces.

Our detailed analysis of Knee 01, which exhibited the most significant trochlear cartilage thickness reduction, provided insight into the potential mechanism that leads to cartilage degeneration following OWHTO. The 3D and 2D MRI analyses revealed an interaction between an inferior patellar pole osteophyte and a trochlear cartilage defect. The postoperative changes in patellar position—specifically the inferior displacement and increased downward tilt—appeared to create a pathological contact between the patellar osteophyte and the trochlear cartilage. Similar contact-related wear was suspected in two additional cases (Knees 04 and 10), in which a medial trochlear osteophyte was likely to have contributed to patellar cartilage loss. Taken together, these findings suggest that bony impingement may be an additional mechanism responsible for cartilage degeneration after OWHTO, particularly in patients with preexisting osteophyte formation.

This study has four main limitations. First, the sample size was small, with only 13 cases, primarily due to the limited availability of both preoperative and postoperative 3D-MRI scan data. Second, 6 of the 13 knees required manual ROI adjustments in the trochlear region and 2 of the 13 knees required manual ROI adjustments in the patellar region, indicating that improvement is needed in the accuracy of automatic ROI setting for knees with deformity. The need for manual adjustment suggests that the current automatic ROI setting may lack precision when used in knees with large cartilage defects or altered morphology. This manual intervention, although it was necessary to ensure correct ROI placement, could introduce some observer dependency. Third, the laterality in PF joint cartilage wear was assessed subjectively to take advantage of the presented 3D images because we considered this to be important for visually conveying the spatial distribution of cartilage changes, rather than relying solely on numerical quantification. Fourth, variations in follow-up timing may have influenced our findings. Although all patients underwent implant removal approximately 1 year after OWHTO, the exact timing varied between the cases. These differences in follow-up duration could affect the extent and pattern of the observed cartilage changes, as cartilage remodeling processes may continue to evolve over time following osteotomy.

Quantitative 3D-MRI analysis precisely revealed the PF joint cartilage thickness changes occurring after OWHTO. Our findings demonstrated that postoperative cartilage thinning occurred predominantly in the medial aspect, with significant thickness decreases observed in both the patellar and trochlear regions. The observed spatial correspondence between

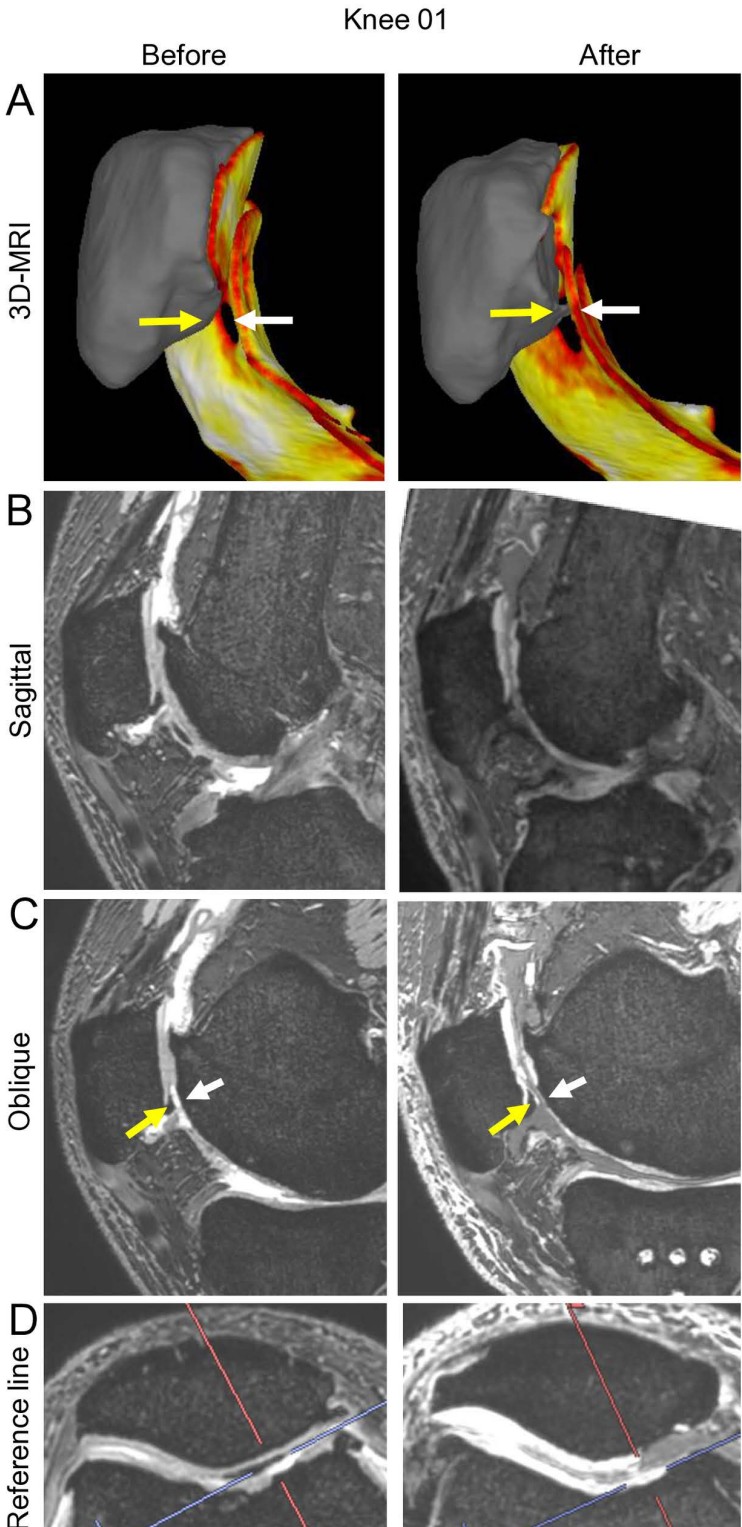

Knee 01

**Fig 7. Detailed analysis of the patellofemoral interactions after OWHTO in Knee 01. (A)** Lateral 3D-MRI view of the patellar bone, patellar cartilage, and femoral trochlear cartilage. Yellow arrow indicates osteophyte formation at the inferior pole of patella. White arrow indicates one of the defects in the femoral trochlear cartilage. **(B)** Sagittal 2D-MRI. Note that, postoperatively, the patella is positioned inferiorly and tilted downward. **(C)** Oblique 2D-MRI.

This slice was selected to demonstrate the relationship between the patellar osteophyte and the trochlear cartilage defect. Note the contact between the osteophyte and the trochlear cartilage defect. **(D)** Reference line for the oblique plane. The red line indicates the slice level of the oblique image in **(C)**, while the blue line shows the perpendicular plane to this slice.

the patellar and trochlear cartilage thinning suggests that degeneration in these opposing surfaces may occur in a coordinated manner, potentially driven by postoperative changes in PF joint alignment and contact mechanics. This spatially matched pattern of cartilage loss, and its likely link to altered biomechanics, provides important insight into how and where PF joint degeneration develops after OWHTO, while highlighting both the specific regions at risk and the mechanisms that may underlie these changes.

The findings presented here enhance our understanding of the biomechanical alterations that occur following OWHTO and may help guide surgical planning and postoperative management. Radiologists should be alert to any medial-dominant PF cartilage thinning and spatially corresponding patellar–trochlear defects evident on postoperative MRI scans, while orthopedic surgeons can use this same information to anticipate PF-related complications and adjust correction targets or rehabilitation protocols accordingly.

Given our small sample size, larger prospective studies are needed to investigate the long-term progression of these cartilage thickness changes and their correlation with clinical outcomes, such as the presence and severity of anterior knee pain, PF joint–specific functional scores (e.g., the Kujala score), return to activity, and the need for further surgical intervention. Also, because our study involved a relatively small number of patients from a single center, any generalizability of our findings should be considered with caution. Validation in larger, more diverse populations is warranted. A better understanding of these factors could ultimately lead to improved treatment strategies for patients undergoing OWHTO.

## Supporting information

**S1 File. Fuchioka dataset 251021.**
(XLSX)

## Acknowledgments

We thank Ms. Ellen Roider for English editing.

## Author contributions

**Conceptualization:** Nobutake Ozeki, Ichiro Sekiya.

**Data curation:** Yusuke Fuchioka, Nobutake Ozeki, Asuka Asami, Ichiro Sekiya.

**Formal analysis:** Yusuke Fuchioka, Nobutake Ozeki, Ichiro Sekiya.

**Funding acquisition:** Hisako Katano, Ichiro Sekiya.

**Investigation:** Yusuke Fuchioka, Nobutake Ozeki, Asuka Asami, Ichiro Sekiya.

**Methodology:** Nobutake Ozeki, Ichiro Sekiya.

**Project administration:** Nobutake Ozeki, Ichiro Sekiya.

**Resources:** Nobutake Ozeki, Hideyuki Koga, Tomomasa Nakamura, Yusuke Nakagawa, Masaki Amemiya, Ichiro Sekiya.

**Supervision:** Nobutake Ozeki, Hideyuki Koga, Hisako Katano, Ichiro Sekiya.

**Validation:** Nobutake Ozeki, Ichiro Sekiya.

**Visualization:** Yusuke Fuchioka, Hisako Katano, Asuka Asami, Ichiro Sekiya.

**Writing – original draft:** Yusuke Fuchioka, Ichiro Sekiya.

**Writing – review & editing:** Yusuke Fuchioka, Hideyuki Koga, Ichiro Sekiya.

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
