## [Decision Letter · Decision Letter 0]

8 Aug 2025

Dear Dr. Sekiya,

We look forward to receiving your revised manuscript.

Kind regards,

Shuyang Han

Academic Editor

PLOS ONE

Journal Requirements:

This study was funded by Institute of Science Tokyo.

The authors declare no competing interests.

5. We note that your Data Availability Statement is currently as follows: All relevant data are within the paper.

7. Please remove all personal information, ensure that the data shared are in accordance with participant consent, and re-upload a fully anonymized data set.

Additional guidance on preparing raw data for publication can be found in our Data Policy (https://journals.plos.org/plosone/s/data-availability#loc-human-research-participant-data-and-other-sensitive-data) and in the following article: http://www.bmj.com/content/340/bmj.c181.long .

Reviewers' comments:

Reviewer's Responses to Questions

**Comments to the Author**

1. Is the manuscript technically sound, and do the data support the conclusions?

Reviewer #1: Yes

2. Has the statistical analysis been performed appropriately and rigorously?

Reviewer #1: Yes

3. Have the authors made all data underlying the findings in their manuscript fully available?

Reviewer #1: Yes

4. Is the manuscript presented in an intelligible fashion and written in standard English?

Reviewer #1: Yes

Reviewer #1: General comments: Congratulations to the authors for performing this insightful study. As a reviewer, I appreciate your scientific efforts in expanding on the topic of this study. The language and grammar used is appropriate and does not require extensive proofreading.

Title:

-Please indicate the study design with a commonly used term.

Abstract:

-Lines 30-36: Check grammar.

-Please provide a hypothesis in the abstract.

-Line 39: What constitutes the term ‘complete’?

-Lines 41-45: Please clarify this information to make it easier for the reader to understand.

-Line 47: Please keep consistency with writing numbers in numerical or worded format.

-Line 48: Please specify the significance level.

-Line 49: Why was 0.1mm chosen as the significant value?

-Lines 51-53: Check grammar.

Introduction: Adequate length and an excellent introduction to the topic covered in the manuscript. Please outline the clinical relevance of the study in more detail – why is study required? What do clinicians hope to gain by reading the results of this study?

-Lines 70-72: Check grammar.

-Line 79: Check grammar.

-Lines 82-83: Check grammar.

-Please add a hypothesis at the end of the introduction section.

Methods: Well thought-out and thorough section, well done to the authors for this robust methodology. Please add a section outlining inclusion and exclusion criteria for patients.

-Line 97: Did all of the patients come from a single center?

-Lines 97-98: Please briefly describe why cases with lateral reticular release were excluded.

-Lines 99-100: Please specify how long before surgery and after implant removal that imaging data was collected. Was there a method of standardization for this?

-Lines 104-105: Why was 57% chosen? Could you please briefly outline this in the manuscript?

-Line 117: Given that cartilage is less accurately measured on radiographs in comparison to MRI, it is important to briefly outline the significance of measurements on radiographs for those without a background in imaging knowledge.

-Lines 118-119: Who and how many people evaluated the images?

-Line 136: Check grammar.

-Lines 146-149: Briefly explain why this was done the way that it was done.

-Line 155: What constituted an ‘inappropriate’ ROI? What problems were commonly encountered?

-Lines 157-158: Please reference this study for the readers wanting more information.

-Lines 163-164: How often was this encountered?

-Line 169: Check grammar.

Results: This is a thorough and well-thought-through section, well done to all of the authors.

-Lines 213-214: Was there anything unique about this patient that could have explained this change?

-Lines 225-226: Again, was there anything unique explaining this change?

-Line 233: Give a brief description on kissing lesions for readers hoping to know more about the topic.

Discussion:

-Lines 256-257: What relationship is being referred to here?

-Lines 271-273: Please move this information to the methods section where it is first referred to.

-Lines 289-290: Check grammar.

-Lines 299-301: Please explain this deduction in more detail – how is this conclusion suggested? Citing other studies with similar results may be useful here.

-Lines 313-316: Did any of the other cases show bony impingement or osteophyte formation which would support or refute this conclusion?

-Lines 319-321: What is the implication of the manual ROI adjustment?

-Lines 323-325: What objective assessment methods should be employed? Were any of these assessment methods ones which the authors could have employed in this study?

-It is important for the limitations section to address the different follow-up times when reviewing imaging studies and the impact of these differences.

-Lines 330-332: What ‘insights’ are being referred to here? Please be specific

-Line 332: This is an excellent and important consideration. What does this mean for radiologists who assess such images? How should it affect their practice? What is the implication for orthopedic surgeons who are performing presurgical planning?

-Lines 337-338: Please outline which clinical outcomes need to be assessed in the future based on the conclusions of the study.

-Please comment on the generalizability of the results in the conclusion section.

References:

-Appropriate.

Figures:

-Adequate and complete.

**Do you want your identity to be public for this peer review?** For information about this choice, including consent withdrawal, please see our Privacy Policy

Reviewer #1: No

---

## [Author Response · Author response to Decision Letter 1]

27 Aug 2025

Reviewer #1: General comments: Congratulations to the authors for performing this insightful study. As a reviewer, I appreciate your scientific efforts in expanding on the topic of this study. The language and grammar used is appropriate and does not require extensive proofreading.

Title:

-Please indicate the study design with a commonly used term.

Thank you for this valuable suggestion. We have revised the title to clearly indicate the study design by adding "a retrospective cohort study" at the end. The revised title now reads: "3D-MRI evaluation of cartilage thickness changes and their location in the patellofemoral joint after open wedge high tibial osteotomy for knee osteoarthritis: a retrospective cohort study".

Abstract:

-Lines 30-36: Check grammar.

The document has been reviewed again by a native speaker.

-Please provide a hypothesis in the abstract.

We have added the following sentence to the Abstract: “We hypothesized that OWHTO would result in measurable decreases in the PF joint cartilage thickness, predominantly medially and detectable using quantitative 3D-MRI.”

-Line 39: What constitutes the term ‘complete’?

The term 'complete' in the original sentence referred to having analyzable 3D-MRI datasets from both required time points (preoperative and post-hardware removal) that were of sufficient quality for cartilage thickness measurement analysis. However, we recognize that this term was vague and could be misinterpreted.

We have rewritten the sentence as follows in the Abstract: “Patients were included if they had undergone OWHTO without lateral retinacular release for medial osteoarthritis and had both the preoperative and post–hardware-removal 3D-MRI datasets required for this analysis.”

We have also rewritten the sentence as follows in the Results: “In total, 13 knees from 13 patients who underwent OWHTO without lateral retinacular release for medial OA between April 2015 and March 2023 also had both the preoperative and post–hardware-removal 3D-MRI datasets required for the present analysis (Figure 2).”

-Lines 41-45: Please clarify this information to make it easier for the reader to understand.

We have rewritten the sentences as follows: “Trochlear and patellar cartilage thicknesses were measured from 3D-MRI images at both time points. Changes exceeding 0.1 mm (the validated measurement precision threshold) were considered significant. To assess cartilage loss location, each 3D image was divided into medial, central, and lateral thirds.”

-Line 47: Please keep consistency with writing numbers in numerical or worded format.

The sentence has been revised to: “In total, 13 knees from 13 patients (median age 55 [32–74] years) were evaluated.” We have also ensured that all other numerical values throughout the manuscript follow the same format."

-Line 48: Please specify the significance level.

We have added “(p<0.001)” as follows: “Postoperatively, patellar height and lateral tilt significantly decreased (p<0.001 for both).”

-Line 49: Why was 0.1mm chosen as the significant value?

We have added the following sentence to the Methods section of the Abstract: “Changes exceeding 0.1 mm (the validated measurement precision threshold) were considered significant.”

-Lines 51-53: Check grammar.

The document has been reviewed again by a native speaker.

Introduction: Adequate length and an excellent introduction to the topic covered in the manuscript. Please outline the clinical relevance of the study in more detail – why is study required? What do clinicians hope to gain by reading the results of this study?

We have added one sentence as follows:

“Accurate quantification of cartilage thickness changes could provide valuable insights into the mechanisms of PF joint degeneration and could potentially guide surgical planning. Understanding these changes could enable surgeons to identify high-risk patients, optimize surgical techniques, and develop targeted rehabilitation protocols to minimize PF joint deterioration. We hypothesized that OWHTO would result in measurable decreases in PF joint cartilage thickness, predominantly in the medial aspect, and that we could detect these changes using quantitative 3D-MRI. The aim of this study was to evaluate the clinical utility of quantitative 3D-MRI in assessing PF joint cartilage changes before and after OWHTO, with particular attention paid to the spatial distribution of cartilage wear and the relationship between patellar and trochlear cartilage changes.”

-Lines 70-72: Check grammar.

The document has been reviewed again by a native speaker.

-Line 79: Check grammar.

The document has been reviewed again by a native speaker.

-Lines 82-83: Check grammar.

The document has been reviewed again by a native speaker.

-Please add a hypothesis at the end of the introduction section.

We have added one sentence as follows:

“Accurate quantification of cartilage thickness changes could provide valuable insights into the mechanisms of PF joint degeneration and could potentially guide surgical planning. Understanding these changes could enable surgeons to identify high-risk patients, optimize surgical techniques, and develop targeted rehabilitation protocols to minimize PF joint deterioration. We hypothesized that OWHTO would result in measurable decreases in PF joint cartilage thickness, predominantly in the medial aspect, and that we could detect these changes using quantitative 3D-MRI. The aim of this study was to evaluate the clinical utility of quantitative 3D-MRI in assessing PF joint cartilage changes before and after OWHTO, with particular attention paid to the spatial distribution of cartilage wear and the relationship between patellar and trochlear cartilage changes.”

Methods: Well thought-out and thorough section, well done to the authors for this robust methodology. Please add a section outlining inclusion and exclusion criteria for patients.

We have added the following section outlining the inclusion and exclusion criteria used when selecting our patients:

“Inclusion criteria:

• Patients aged 18 years or older

• Diagnosis of medial compartment knee OA

• Underwent OWHTO without lateral retinacular release of the PF joint

• 3D-MRI data available both preoperatively and after implant removal

• Minimum follow-up period of 1 year

Exclusion criteria:

• Previous knee surgery on the affected limb

• Incomplete or poor-quality MRI data, loss to follow-up”

-Line 97: Did all of the patients come from a single center?

We have added the following information: “We retrospectively included patients from a single center who underwent opening wedge high tibial osteotomy (OWHTO) without lateral retinacular release of the PF joint for medial OA between 2015 and 2023. Cases with lateral retinacular release were excluded because the procedure can alter PF joint alignment and cartilage status. All included patients had analyzable 3D-MRI data obtained following a standardized imaging protocol within 3 months before surgery and within 3 months after implant removal.”

-Lines 97-98: Please briefly describe why cases with lateral reticular release were excluded.

We have added the following sentence: “Cases with lateral retinacular release were excluded because the procedure can alter PF joint alignment and cartilage status.”

-Lines 99-100: Please specify how long before surgery and after implant removal that imaging data was collected. Was there a method of standardization for this?

We have added the following information: “All included patients had analyzable 3D-MRI data obtained following a standardized imaging protocol within 3 months before surgery and within 3 months after implant removal.”

-Lines 104-105: Why was 57% chosen? Could you please briefly outline this in the manuscript?

We have added the following sentence: “The OWHTO procedure followed the method proposed by Staubli et al. [11]. The aim of the preoperative plan was to shift the weight-bearing line ratio to a point 57% laterally along the transverse diameter of the tibial plateau. This target was chosen based on the method described by Katagiri et al., in which centralization and preservation of the medial meniscus function lead to a correction closer to neutral alignment than is achieved using the conventional 62% setting [12].”

-Line 117: Given that cartilage is less accurately measured on radiographs in comparison to MRI, it is important to briefly outline the significance of measurements on radiographs for those without a background in imaging knowledge.

We have added the following sentences: “Although cartilage thickness cannot be directly assessed on radiographs with the same accuracy as it can on MRI, these radiographic measurements provide valuable information on lower limb alignment, patellar position, and other bony parameters that can influence PF joint mechanics. These parameters are essential for interpreting postoperative changes in cartilage condition in the context of overall knee biomechanics.”

-Lines 118-119: Who and how many people evaluated the images?

We have added the following sentence: “All radiographic measurements were performed by a single experienced orthopedic surgeon (YF), and in cases of uncertainty, the final decision was made in consultation with another senior orthopedic surgeon (NO).”

-Line 136: Check grammar.

The document has been reviewed again by a native speaker.

-Lines 146-149: Briefly explain why this was done the way that it was done.

We have added the following sentence: “This orientation was chosen because it provides the clearest view and the most comprehensive information for assessment, while also standardizing the measurements and reducing variability.”

-Line 155: What constituted an ‘inappropriate’ ROI? What problems were commonly encountered?

We have added the following sentence: “When the preoperative ROI was deemed inappropriate, manual corrections were made, and these cases were documented. An ROI was considered inappropriate when it was clearly misaligned with the actual femoral trochlea or patellar cartilage; this most commonly occurred in cases with large cartilage defects and markedly altered cartilage morphology.”

-Lines 157-158: Please reference this study for the readers wanting more information.

We have added the following information: “Based on the analytical study by Katano et al., who examined inter-measurement errors of cartilage thickness, cases with cartilage thickness changes exceeding 0.1 mm were considered to be beyond the measurement error [17].” This reference has been published recently in PLOS ONE.

-Lines 163-164: How often was this encountered?

We have added the following information: “The 3D images were then visually assessed to determine which of these three subregions had changed. This visual assessment was performed for all cases.”

-Line 169: Check grammar.

The document has been reviewed again by a native speaker.

Results: This is a thorough and well-thought-through section, well done to all of the authors.

-Lines 213-214: Was there anything unique about this patient that could have explained this change?

We carefully examined the MRI and radiographs but could not find anything unique. We have added the following sentence: “One case (Knee 13) showed an increase exceeding 0.1 mm, with changes observed in the lateral 1/3. A review of this patient’s MRI and radiographic data revealed no specific factors that could explain this change.”

-Lines 225-226: Again, was there anything unique explaining this change?

We carefully examined the MRI and radiographs for this case but could not find anything unique. We have added the following sentence: “One case (Knee 08) showed an increase exceeding 0.1 mm, with changes apparent throughout the entire cartilage. No distinctive imaging findings were identified that could account for this observation.”

-Line 233: Give a brief description on kissing lesions for readers hoping to know more about the topic.

We have added the following explanation: “These matching defects represented kissing lesions, which refer to opposing cartilage defects on articulating surfaces [18].”

Discussion:

-Lines 256-257: What relationship is being referred to here?

We have added the following information: “Some cases showed spatial correspondence between patellar and trochlear cartilage defects, suggesting the possibility that these opposing defects may develop through mechanical interaction, similar to "kissing lesions" [18].”

-Lines 271-273: Please move this information to the methods section where it is first referred to.

We have already included the following statement in the Methods section: “Based on the analytical study by Katano et al., who examined inter-measurement errors of cartilage thickness, cases with cartilage thickness changes exceeding 0.1 mm were considered to be beyond the measurement error [17].” However, we believe that retaining this paragraph in the Discussion is important, as it provides the detailed validation data supporting the selection of the 0.1 mm threshold, which is critical for interpreting our results.

-Lines 289-290: Check grammar.

The document has been reviewed again by a native speaker.

-Lines 299-301: Please explain this deduction in more detail – how is this conclusion suggested? Citing other studies with similar results may be useful here.

We have rewritten the paragraph as follows:

“Our spatial analysis of cartilage defects using superimposed 3D MRI images provided valuable insights into the pattern of PF joint degeneration after OWHTO. Of particular interest was the observation of kissing lesions, where the cartilage defects on the patellar surface corresponded spatially with the defects on the trochlear surface in two of the three cases that showed postoperative patellar cartilage defects. The presence of these kissing lesions strongly suggests that the cartilage degeneration follows a coordinated pattern of wear, likely resulting from the altered contact mechanics after OWHTO. This deduction is supported by the spatial correspondence of patellar and trochlear defects observed in multiple cases, indicating that degeneration on one surface occurs in regions directly opposing degeneration on the other. Similar matched defect patterns in the PF joint have been reported in studies of malalignment and altered PF kinematics, where increased localized contact stress was associated with accelerated cartilage loss [25, 26]. These findings collectively support the interpretation that postoperative changes in PF alignment and biomechanics after OWHTO can create specific zones of elevated mechanical stress that can lead to the coordinated degeneration of opposing cartilage surfaces.”

-Lines 313-316: Did any of the other cases show bony impingement or osteophyte formation which would support or refute this conclusion?

We have added following information to the Results:

“Patellar cartilage superimposed on trochlear cartilage

Three cases (Knees 01, 04, and 10) showed postoperative patellar cartilage defects in the 3D MRI images when the patellar cartilage was superimposed on the trochlear cartilage (Figure 6). In two of these cases (Knees 01 and 04), the locations of cartilage defects matched between the patellar and trochlear surfaces. These matching defects represented kissing lesions, which refer to opposing cartilage defects on articulating surfaces [18]. In all three cases, the findings suggest that contact with a medial trochlear osteophyte may have contributed to patellar cartilage wear.”

We have also rewritten the paragraph in the Discussion as follows:

“Our detailed analysis of Knee 01, which exhibited the most significant trochlear cartilage thickness reduction, provided insight into the potential mechanism that leads to cartilage degeneration following OWHTO. The 3D and 2D MRI analyses revealed an interaction between an inferior patellar pole osteophyte and a trochlear cartilage defect. The postoperative changes in patellar position—specifically the inferior displacement and increased downward tilt—appeared to create a pathological contact between the patellar osteoph

---

## [Decision Letter · Decision Letter 1]

21 Sep 2025

3D-MRI evaluation of cartilage thickness changes and their location in the patellofemoral joint after open wedge high tibial osteotomy for knee osteoarthritis: a retrospective cohort study

PONE-D-25-19799R1

Dear Dr. Sekiya,

We’re pleased to inform you that your manuscript has been judged scientifically suitable for publication and will be formally accepted for publication once it meets all outstanding technical requirements.

Kind regards,

Shuyang Han

Academic Editor

PLOS ONE

Additional Editor Comments (optional):

Reviewer #1:

Reviewers' comments:

Reviewer's Responses to Questions

**Comments to the Author**

Reviewer #1: All comments have been addressed

2. Is the manuscript technically sound, and do the data support the conclusions?

Reviewer #1: Yes

3. Has the statistical analysis been performed appropriately and rigorously?

Reviewer #1: Yes

4. Have the authors made all data underlying the findings in their manuscript fully available?

Reviewer #1: Yes

5. Is the manuscript presented in an intelligible fashion and written in standard English?

Reviewer #1: Yes

Reviewer #1: (No Response)

**Do you want your identity to be public for this peer review?** For information about this choice, including consent withdrawal, please see our Privacy Policy

Reviewer #1: No

---

## [Editor Report · Acceptance letter]

PONE-D-25-19799R1

PLOS ONE

Dear Dr. Sekiya,

I'm pleased to inform you that your manuscript has been deemed suitable for publication in PLOS ONE. Congratulations! Your manuscript is now being handed over to our production team.

Kind regards,

on behalf of

Dr. Shuyang Han

Academic Editor

PLOS ONE